# Deep ADMM-Net for Compressive Sensing MRI

**Yan Yang**
Xi'an Jiaotong University
yangyan92@stu.xjtu.edu.cn

**Jian Sun**
Xi'an Jiaotong University
jiansun@mail.xjtu.edu.cn

**Huibin Li**
Xi'an Jiaotong University
huibinli@mail.xjtu.edu.cn

**Zongben Xu**
Xi'an Jiaotong University
zbxu@mail.xjtu.edu.cn

## Abstract

Compressive Sensing (CS) is an effective approach for fast Magnetic Resonance Imaging (MRI). It aims at reconstructing MR image from a small number of under-sampled data in $k$-space, and accelerating the data acquisition in MRI. To improve the current MRI system in reconstruction accuracy and computational speed, in this paper, we propose a novel deep architecture, dubbed ADMM-Net. ADMM-Net is defined over a data flow graph, which is derived from the iterative procedures in Alternating Direction Method of Multipliers (ADMM) algorithm for optimizing a CS-based MRI model. In the training phase, all parameters of the net, e.g., image transforms, shrinkage functions, etc., are discriminatively trained end-to-end using L-BFGS algorithm. In the testing phase, it has computational overhead similar to ADMM but uses optimized parameters learned from the training data for CS-based reconstruction task. Experiments on MRI image reconstruction under different sampling ratios in $k$-space demonstrate that it significantly improves the baseline ADMM algorithm and achieves high reconstruction accuracies with fast computational speed.

## 1 Introduction

Magnetic Resonance Imaging (MRI) is a non-invasive imaging technique providing both functional and anatomical information for clinical diagnosis. Imaging speed is a fundamental challenge. Fast MRI techniques are essentially demanded for accelerating data acquisition while still reconstructing high quality image. Compressive sensing MRI (CS-MRI) is an effective approach allowing for data sampling rate much lower than Nyquist rate without significantly degrading the image quality [1].

CS-MRI methods first sample data in $k$-space (i.e., Fourier space), then reconstruct image using compressive sensing theory. Regularization related to the data prior is a key component in a CS-MRI model to reduce imaging artifacts and improve imaging precision. Sparse regularization can be explored in specific transform domain or general dictionary-based subspace [2]. Total Variation (TV) regularization in gradient domain has been widely utilized in MRI [3, 4]. Although it is easy and fast to optimize, it introduces staircase artifacts in reconstructed image. Methods in [5, 6] leverage sparse regularization in the wavelet domain. Dictionary learning methods rely on a dictionary of local patches to improve the reconstruction accuracy [7, 8]. The non-local method uses groups of similar local patches for joint patch-level reconstruction to better preserve image details [9, 10, 11]. In performance, the basic CS-MRI methods run fast but produce less accurate reconstruction results. The non-local and dictionary learning-based methods generally output higher quality MR images, but suffer from slow reconstruction speed. In a CS-MRI model, it is commonly challenging to choose an optimal image transform domain / subspace and the corresponding sparse regularization.

To optimize the CS-MRI models, Alternating Direction Method of Multipliers (ADMM) has proven to be an efficient variable splitting algorithm with convergence guarantee [4, 12, 13]. It considers the augmented Lagrangian function of a given CS-MRI model, and splits variables into subgroups, which can be alternatively optimized by solving a few simply subproblems. Although ADMM is generally efficient, it is not trivial to determine the optimal parameters (e.g., update rates, penalty parameters) influencing accuracy in CS-MRI.

In this work, we aim to design a fast yet accurate method to reconstruct high-quality MR images from under-sampled $k$-space data. We propose a novel deep architecture, dubbed *ADMM-Net*, inspired by the ADMM iterative procedures for optimizing a general CS-MRI model. This deep architecture consists of multiple stages, each of which corresponds to an iteration in ADMM algorithm. More specifically, we define a deep architecture represented by a data flow graph [14] for ADMM procedures. The operations in ADMM are represented as graph nodes, and the data flow between two operations in ADMM is represented by a directed edge. Therefore, the ADMM iterative procedures naturally determine a deep architecture over a data flow graph. Given an under-sampled data in $k$-space, it flows over the graph and generates a reconstructed image. All the parameters (e.g., transforms, shrinkage functions, penalty parameters, etc.) in the deep architecture can be discriminatively learned from training pairs of under-sampled data in $k$-space and reconstructed image using fully sampled data by backpropagation [15] over the data flow graph.

Our experiments demonstrate that the proposed deep ADMM-Net is effective both in reconstruction accuracy and speed. Compared with the baseline methods using sparse regularization in transform domain, it achieves significantly higher accuracy and takes comparable computational time. Compared with the state-of-the-art methods using dictionary learning and non-local techniques, it achieves high accuracy in significantly faster computational speed.

The main contributions of this paper can be summarized as follows. We propose a novel deep ADMM-Net by reformulating an ADMM algorithm to a deep network for CS-MRI. This is achieved by designing a data flow graph for ADMM to effectively build and train the ADMM-Net. ADMM-Net achieves high accuracy in MR image reconstruction with fast computational speed justified in experiments. The discriminative parameter learning approach has been applied to sparse coding and Markov Random Filed [16, 17, 18, 19]. But, to the best of our knowledge, this is the first computational framework that maps an ADMM algorithm to a learnable deep architecture.

## 2   Deep ADMM-Net for Fast MRI

### 2.1   Compressive Sensing MRI Model and ADMM Algorithm

**General CS-MRI Model**: Assume $x \in \mathbb{C}^N$ is an MRI image to be reconstructed, $y \in \mathbb{C}^{N'}$ ($N' < N$) is the under-sampled $k$-space data, according to the CS theory, the reconstructed image can be estimated by solving the following optimization problem:

$$\hat{x} = \arg\min_x \left\{ \frac{1}{2} \|Ax - y\|_2^2 + \sum_{l=1}^L \lambda_l g(D_l x) \right\}, \tag{1}$$

where $A = PF \in \mathbb{R}^{N' \times N}$ is a measurement matrix, $P \in \mathbb{R}^{N' \times N}$ is a under-sampling matrix, and $F$ is a Fourier transform. $D_l$ denotes a transform matrix for a filtering operation, e.g., Discrete Wavelet Transform (DWT), Discrete Cosine Transform (DCT), etc. $g(\cdot)$ is a regularization function derived from the data prior, e.g., $l_q$-norm ($0 \leq q \leq 1$) for a sparse prior. $\lambda_l$ is a regularization parameter.

**ADMM solver**: [12] The above optimization problem can be solved efficiently using ADMM algorithm. By introducing auxiliary variables $z = \{z_1, z_2, \cdots, z_L\}$, Eqn. (1) is equivalent to:

$$\min_{x,z} \frac{1}{2} \|Ax - y\|_2^2 + \sum_{l=1}^L \lambda_l g(z_l) \quad s.t. \ z_l = D_l x, \ \forall \, l \in [1, 2, \cdots, L]. \tag{2}$$

Its augmented Lagrangian function is :

$$\mathcal{L}_\rho(x, z, \alpha) = \frac{1}{2} \|Ax - y\|_2^2 + \sum_{l=1}^L \lambda_l g(z_l) - \sum_{l=1}^L \langle \alpha_l, z_l - D_l x \rangle + \sum_{l=1}^L \frac{\rho_l}{2} \|z_l - D_l x\|_2^2, \tag{3}$$

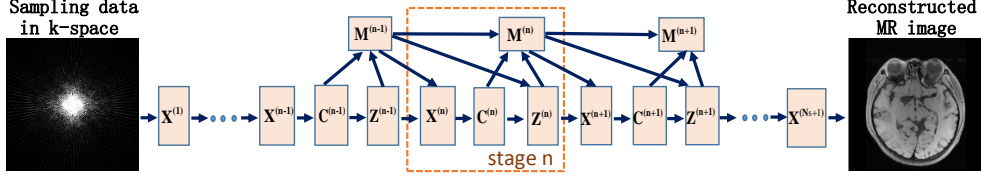

Figure 1: The data flow graph for the ADMM optimization of a general CS-MRI model. This graph consists of four types of nodes: reconstruction ($\mathbf{X}$), convolution ($\mathbf{C}$), non-linear transform ($\mathbf{Z}$), and multiplier update ($\mathbf{M}$). An under-sampled data in $k$-space is successively processed over the graph, and finally generates a MR image. Our deep ADMM-Net is defined over this data flow graph.

where $\alpha = \{\alpha_l\}$ are Lagrangian multipliers and $\rho = \{\rho_l\}$ are penalty parameters. ADMM alternatively optimizes $\{x, z, \alpha\}$ by solving the following three subproblems:

$$\begin{cases} x^{(n+1)} = \underset{x}{\arg\min} \frac{1}{2}\|Ax - y\|_2^2 - \sum_{l=1}^{L}\langle \alpha_l^{(n)}, z_l^{(n)} - D_l x\rangle + \sum_{l=1}^{L} \frac{\rho_l}{2}\|z_l^{(n)} - D_l x\|_2^2, \\ z^{(n+1)} = \underset{z}{\arg\min} \sum_{l=1}^{L} \lambda_l g(z_l) - \sum_{l=1}^{L}\langle \alpha_l^{(n)}, z_l - D_l x^{(n+1)}\rangle + \sum_{l=1}^{L} \frac{\rho_l}{2}\|z_l - D_l x^{(n+1)}\|_2^2, \\ \alpha^{(n+1)} = \underset{\alpha}{\arg\min} \sum_{l=1}^{L}\langle \alpha_l, D_l x^{(n+1)} - z_l^{(n+1)}\rangle, \end{cases}$$

(4)

where $n \in [1, 2, \cdots, N_s]$ denotes $n$-th iteration. For simplicity, let $\beta_l = \frac{\alpha_l}{\rho_l}$ ($l \in [1, 2, \cdots, L]$), and substitute $A = PF$ into Eqn. (4). Then the three subproblems have the following solutions:

$$\begin{cases} \mathbf{X}^{(\mathbf{n})} : x^{(n)} = F^T[P^TP + \sum_{l=1}^{L}\rho_l F D_l^T D_l F^T]^{-1}[P^T y + \sum_{l=1}^{L}\rho_l F D_l^T(z_l^{(n-1)} - \beta_l^{(n-1)})], \\ \mathbf{Z}^{(\mathbf{n})} : z_l^{(n)} = S(D_l x^{(n)} + \beta_l^{(n-1)}; \lambda_l/\rho_l), \\ \mathbf{M}^{(\mathbf{n})} : \beta_l^{(n)} = \beta_l^{(n-1)} + \eta_l(D_l x^{(n)} - z_l^{(n)}), \end{cases}$$

(5)

where $x^{(n)}$ can be efficiently computed by fast Fourier transform, $S(\cdot)$ is a nonlinear shrinkage function. It is usually a soft or hard thresholding function corresponding to the sparse regularization of $l_1$-norm and $l_0$-norm respectively [20]. The parameter $\eta_l$ is an update rate.

In CS-MRI, it commonly needs to run the ADMM algorithm in dozens of iterations to get a satisfactory reconstruction result. However, it is challenging to choose the transform $D_l$ and shrinkage function $S(\cdot)$ for general regularization function $g(\cdot)$. Moreover, it is also not trivial to tune the parameters $\rho_l$ and $\eta_l$ for $k$-space data with different sampling ratios. To overcome these difficulties, we will design a data flow graph for the ADMM algorithm, over which we can define a deep ADMM-Net to discriminatively learn all the above transforms, functions, and parameters.

## 2.2 Data Flow Graph for the ADMM Algorithm

To design our deep ADMM-Net, we first map the ADMM iterative procedures in Eqn. (5) to a data flow graph [14]. As shown in Fig. 1, this graph comprises of *nodes* corresponding to different operations in ADMM, and *directed edges* corresponding to the data flows between operations. In this case, the $n$-th iteration of ADMM algorithm corresponds to the $n$-th stage of the data flow graph. In the $n$-th stage of the graph, there are four types of nodes mapped from four types of operations in ADMM, i.e., reconstruction operation ($\mathbf{X}^{(\mathbf{n})}$), convolution operation ($\mathbf{C}^{(\mathbf{n})}$) defined by $\{D_l x^{(n)}\}_{l=1}^{L}$, nonlinear transform operation ($\mathbf{Z}^{(\mathbf{n})}$) defined by $S(\cdot)$, and multiplier update operation ($\mathbf{M}^{(\mathbf{n})}$) in Eqn. (5). The whole data flow graph is a multiple repetition of the above stages corresponding to successive iterations in ADMM. Given an under-sampled data in $k$-space, it flows over the graph and finally generates a reconstructed image. In this way, we map the ADMM iterations to a data flow graph, which is useful to define and train our deep ADMM-Net in the following sections.

## 2.3 Deep ADMM-Net

Our deep ADMM-Net is defined over the data flow graph. It keeps the graph structure but generalizes the four types of operations to have learnable parameters as network layers. These operations are now generalized as reconstruction layer, convolution layer, non-linear transform layer, and multiplier update layer. We next discuss them in details.

**Reconstruction layer** ($\mathbf{X^{(n)}}$): This layer reconstructs an MRI image following the reconstruction operation $\mathbf{X^{(n)}}$ in Eqn. (5). Given $z_l^{(n-1)}$ and $\beta_l^{(n-1)}$, the output of this layer is defined as:

$$x^{(n)} = F^T(P^TP + \sum_{l=1}^{L}\rho_l^{(n)}FH_l^{(n)T}H_l^{(n)}F^T)^{-1}[P^Ty + \sum_{l=1}^{L}\rho_l^{(n)}FH_l^{(n)T}(z_l^{(n-1)} - \beta_l^{(n-1)})], \quad (6)$$

where $H_l^{(n)}$ is the $l$-th filter, $\rho_l^{(n)}$ is the $l$-th penalty parameter, $l = 1, \cdots, L$, and $y$ is the input under-sampled data in $k$-space. In the first stage ($n = 1$), $z_l^{(0)}$ and $\beta_l^{(0)}$ are initialized to zeros, therefore $x^{(1)} = F^T(P^TP + \sum_{l=1}^{L}\rho_l^{(1)}FH_l^{(1)T}H_l^{(1)}F^T)^{-1}(P^Ty)$.

**Convolution layer** ($\mathbf{C^{(n)}}$): It performs convolution operation to transform an image into transform domain. Given an image $x^{(n)}$, i.e., a reconstructed image in stage $n$, the output is

$$c_l^{(n)} = D_l^{(n)}x^{(n)}, \quad (7)$$

where $D_l^{(n)}$ is a learnable filter matrix in stage $n$. Different from the original ADMM, we do not constrain the filters $D_l^{(n)}$ and $H_l^{(n)}$ to be the same to increase the network capacity.

**Nonlinear transform layer** ($\mathbf{Z^{(n)}}$): This layer performs nonlinear transform inspired by the shrinkage function $S(\cdot)$ defined in $\mathbf{Z^{(n)}}$ in Eqn. (5). Instead of setting it to be a shrinkage function determined by the regularization term $g(\cdot)$ in Eqn. (1), we aim to learn more general function using piecewise linear function. Given $c_l^{(n)}$ and $\beta_l^{(n-1)}$, the output of this layer is defined as:

$$z_l^{(n)} = S_{PLF}(c_l^{(n)} + \beta_l^{(n-1)}; \{p_i, q_{l,i}^{(n)}\}_{i=1}^{N_c}), \quad (8)$$

where $S_{PLF}(\cdot)$ is a piecewise linear function determined by a set of control points $\{p_i, q_{l,i}^{(n)}\}_{i=1}^{N_c}$. i.e.

$$S_{PLF}(a; \{p_i, q_{l,i}^{(n)}\}_{i=1}^{N_c}) = \begin{cases} a + q_{l,1}^{(n)} - p_1, & a < p_1, \\ a + q_{l,N_c}^{(n)} - p_{N_c}, & a > p_{N_c}, \\ q_{l,k}^{(n)} + \frac{(a-p_k)(q_{l,k+1}^{(n)} - q_{l,k}^{(n)})}{p_{k+1} - p_k}, & p_1 \leq a \leq p_{N_c}, \end{cases} \quad (9)$$

where $k = \lfloor\frac{a-p_1}{p_2-p_1}\rfloor$, $\{p_i\}_{i=1}^{N_c}$ are predefined positions uniformly located within [-1,1], and $\{q_{l,i}^{(n)}\}_{i=1}^{N_c}$ are the values at these positions for $l$-th filter in $n$-th stage. Figure 2 gives an illustrative example. Since a piecewise linear function can approximate any function, we can learn flexible nonlinear transform function from data beyond the off-the-shelf hard or soft thresholding functions.

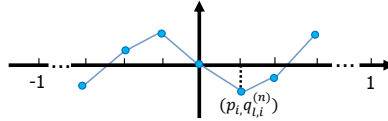

Figure 2: Illustration of a piecewise linear function determined by a set of control points.

**Multiplier update layer** ($\mathbf{M^{(n)}}$): This layer is defined by the Lagrangian multiplier updating procedure $\mathbf{M^{(n)}}$ in Eqn. (5). The output of this layer in stage $n$ is defined as:

$$\beta_l^{(n)} = \beta_l^{(n-1)} + \eta_l^{(n)}(c_l^{(n)} - z_l^{(n)}), \quad (10)$$

where $\eta_l^{(n)}$ are learnable parameters.

**Network Parameters**: These layers are organized in a data flow graph shown in Fig. 1. In the deep architecture, we aim to learn the following parameters: $H_l^{(n)}$ and $\rho_l^{(n)}$ in reconstruction layer, filters $D_l^{(n)}$ in convolution layer, $\{q_{l,i}^{(n)}\}_{i=1}^{N_c}$ in nonlinear transform layer, $\eta_l^{(n)}$ in multiplier update layer, where $l \in [1, 2, \cdots, L]$ and $n \in [1, 2, \cdots, N_s]$ are the indexes for the filters and stages respectively. All of these parameters are taken as the network parameters to be learned.

Figure 3 shows an example of a deep ADMM-Net with three stages. The under-sampled data in $k$-space flows over three stages in a order from circled number 1 to number 12, followed by a final reconstruction layer with circled number 13 and generates a reconstructed image. Immediate reconstruction result at each stage is shown under each reconstruction layer.

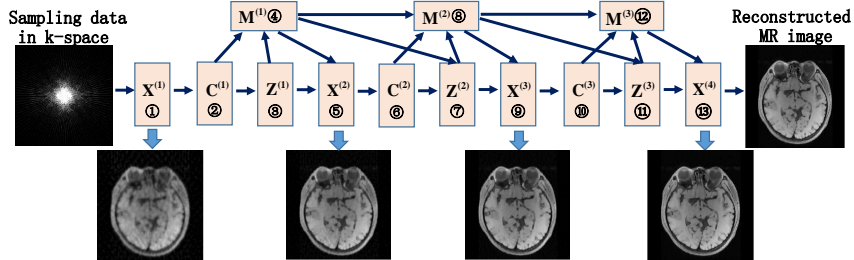

Figure 3: An example of deep ADMM-Net with three stages. The sampled data in $k$-space is successively processed by operations in a order from 1 to 12, followed by a reconstruction layer $X^{(4)}$ to output the final reconstructed image. The reconstructed image in each stage is shown under each reconstruction layer.

# 3 Network Training

We take the reconstructed MR image using fully sampled data in $k$-space as the ground-truth MR image $x^{gt}$, and under-sampled data $y$ in $k$-space as the input. Then a training set $\Gamma$ is constructed containing pairs of under-sampled data and ground-truth MR image. We choose normalized mean square error (NMSE) as the loss function in network training. Given pairs of training data, the loss between the network output and ground truth is defined as:

$$E(\Theta) = \frac{1}{|\Gamma|} \sum_{(y,x^{gt}) \in \Gamma} \frac{\sqrt{\|\hat{x}(y,\Theta) - x^{gt}\|_2^2}}{\sqrt{\|x^{gt}\|_2^2}}, \tag{11}$$

where $\hat{x}(y,\Theta)$ is the network output based on network parameter $\Theta$ and under-sampled data $y$ in $k$-space. We learn the parameters $\Theta = \{(q_{l,i}^{(n)})_{i=1}^{N_c}, D_l^{(n)}, H_l^{(n)}, \rho_l^{(n)}, \eta_l^{(n)}\}_{n=1}^{N_s} \cup \{H_l^{(N_s+1)}, \rho_l^{(N_s+1)}\}$ $(l = 1, \cdots, L)$ by minimizing the loss w.r.t. them using L-BFGS[1]. In the following, we first discuss the initialization of these parameters and then compute the gradients of the loss function $E(\Theta)$ w.r.t. parameters $\Theta$ using backpropagation (BP) [21] over the data flow graph.

## 3.1 Initialization

We initialize the network parameters $\Theta$ according to the ADMM solver of the following baseline CS-MRI model:

$$\arg\min_x \left\{ \frac{1}{2}\|Ax - y\|_2^2 + \lambda \sum_{l=1}^{L} \|D_l x\|_1 \right\}. \tag{12}$$

In this model, we set $D_l$ as a DCT basis and impose $l_1$-norm regularization in the DCT transform space. The function $S(\cdot)$ in ADMM algorithm (Eqn. (5)) is a soft thresholding function: $S(t; \lambda/\rho_l) = sgn(t)(|t| - \lambda/\rho_l)$ when $|t| > \lambda/\rho_l$, and 0 otherwise. For each $n$-th stage of deep ADMM-Net, filters $D_l^{(n)}$ in convolution layers and $H_l^{(n)}$ in reconstruction layers are initialized to be $D_l$ in Eqn. (12). In the nonlinear transform layer, we uniformly choose 101 positions located within [-1,1], and each value $q_{l,i}^{(n)}$ is initialized as $S(p_i; \lambda/\rho_l)$. Parameters $\lambda, \rho_l^{(n)}, \eta_l^{(n)}$ are initialized to be the corresponding values in the ADMM algorithm. In this case, the initialized net is exactly a realization of ADMM optimizing Eqn. (12), therefore outputs the same reconstructed image as the ADMM algorithm. The optimization of the network parameters is expected to produce improved reconstruction result.

## 3.2 Gradient Computation by Backpropagation over Data Flow Graph

It is challenging to compute the gradients of loss w.r.t. parameters using backpropagation over the deep architecture in Fig. 1, because it is a directed graph. In the forward pass, we process the data of $n$-th stage in the order of $\mathbf{X^{(n)}}, \mathbf{C^{(n)}}, \mathbf{Z^{(n)}}$ and $\mathbf{M^{(n)}}$. In the backward pass, the gradients are

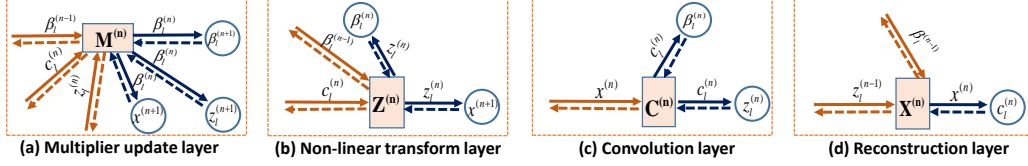

**(a) Multiplier update layer**   **(b) Non-linear transform layer**   **(c) Convolution layer**   **(d) Reconstruction layer**

Figure 4: Illustration of four types of graph nodes (i.e., layers in network) and their data flows in stage $n$. The solid arrow indicates the data flow in forward pass and dashed arrow indicates the backward pass when computing gradients in backpropagation.

computed in an inverse order. Figure 3 shows an example, where the gradient can be computed backwardly from the layers with circled number 13 to 1 successively. For a stage $n$, Fig. 4 shows four types of nodes (i.e., network layers) and the data flow over them. Each node has multiple inputs and (or) outputs. We next briefly introduce the gradients computation for each layer in a typical stage $n$ ($n < N_s$). Please refer to supplementary material for details.

**Multiplier update layer ($\mathbf{M^{(n)}}$):** As shown in Fig. 4(a), this layer has three sets of inputs: $\{\beta_l^{(n-1)}\}, \{c_l^{(n)}\}$ and $\{z_l^{(n)}\}$. Its output $\{\beta_l^{(n)}\}$ is the input to compute $\{\beta_l^{(n+1)}\}, \{z_l^{(n+1)}\}$ and $x^{(n+1)}$. The parameters of this layer are $\eta_l^{(n)}, l = 1, \cdots, L$. The gradients of loss w.r.t. the parameters can be computed as:

$$\frac{\partial E}{\partial \eta_l^{(n)}} = \frac{\partial E}{\partial \beta_l^{(n)}} \frac{\partial \beta_l^{(n)}}{\partial \eta_l^{(n)}}, \text{ where } \frac{\partial E}{\partial \beta_l^{(n)}} = \frac{\partial E}{\partial \beta_l^{(n+1)}} \frac{\partial \beta_l^{(n+1)}}{\partial \beta_l^{(n)}} + \frac{\partial E}{\partial z_l^{(n+1)}} \frac{\partial z_l^{(n+1)}}{\partial \beta_l^{(n)}} + \frac{\partial E}{\partial x^{(n+1)}} \frac{\partial x^{(n+1)}}{\partial \beta_l^{(n)}}.$$

$\frac{\partial E}{\partial \beta_l^{(n)}}$ is the summation of gradients along the three dashed blue arrows in Fig. 4(a). We also compute gradients of the output in this layer w.r.t. its inputs: $\frac{\partial \beta_l^{(n)}}{\partial \beta_l^{(n-1)}}, \frac{\partial \beta_l^{(n)}}{\partial c_l^{(n)}}$, and $\frac{\partial \beta_l^{(n)}}{\partial z_l^{(n)}}$.

**Nonlinear transform layer ($\mathbf{Z^{(n)}}$):** As shown in Fig. 4(b), this layer has two sets of inputs: $\{\beta_l^{(n-1)}\}, \{c_l^{(n)}\}$, and its output $\{z_l^{(n)}\}$ is the input for computing $\{\beta_l^{(n)}\}$ and $x^{(n+1)}$ in next stage. The parameters of this layers are $\{q_{l,i}^{(n)}\}_{i=1}^{N_c}, l = 1, \cdots, L$. The gradient of loss w.r.t. parameters can be computed as

$$\frac{\partial E}{\partial q_{l,i}^{(n)}} = \frac{\partial E}{\partial z_l^{(n)}} \frac{\partial z_l^{(n)}}{\partial q_{l,i}^{(n)}}, \text{where } \frac{\partial E}{\partial z_l^{(n)}} = \frac{\partial E}{\partial \beta_l^{(n)}} \frac{\partial \beta_l^{(n)}}{\partial z_l^{(n)}} + \frac{\partial E}{\partial x^{(n+1)}} \frac{\partial x^{(n+1)}}{\partial z_l^{(n)}}.$$

We also compute the gradients of layer output to its inputs: $\frac{\partial z_l^{(n)}}{\partial \beta_l^{(n)}}$ and $\frac{\partial z_l^{(n)}}{\partial c_l^{(n)}}$.

**Convolution layer ($\mathbf{C^{(n)}}$):** The parameters of this layer are $D_l^{(n)}$ ($l = 1, \cdots, L$). We represent the filter by $D_l^{(n)} = \sum_{m=1}^{t} \omega_{l,m}^{(n)} B_m$, where $B_m$ is a basis element, and $\{\omega_{l,m}^{(n)}\}$ is the set of filter coefficients to be learned. The gradients of loss w.r.t. filter coefficients are computed as

$$\frac{\partial E}{\partial \omega_{l,m}^{(n)}} = \frac{\partial E}{\partial c_l^{(n)}} \frac{\partial c_l^{(n)}}{\partial \omega_{l,m}^{(n)}}, \text{where } \frac{\partial E}{\partial c_l^{(n)}} = \frac{\partial E}{\partial \beta_l^{(n)}} \frac{\partial \beta_l^{(n)}}{\partial c_l^{(n)}} + \frac{\partial E}{\partial z_l^{(n)}} \frac{\partial z_l^{(n)}}{\partial c_l^{(n)}}.$$

The gradient of layer output w.r.t. input is computed as $\frac{\partial c_l^{(n)}}{\partial x^{(n)}}$.

**Reconstruction layer ($\mathbf{X^{(n)}}$):** The parameters of this layer are $H_l^{(n)}, \rho_l^{(n)}$ ($l = 1, \cdots, L$). Similar to convolution layer, we represent the filter by $H_l^{(n)} = \sum_{m=1}^{s} \gamma_{l,m}^{(n)} B_m$, where $\{\gamma_{l,m}^{(n)}\}$ is the set of filter coefficients to be learned. The gradients of loss w.r.t. parameters are computed as

$$\frac{\partial E}{\partial \gamma_{l,m}^{(n)}} = \frac{\partial E}{\partial x^{(n)}} \frac{\partial x^{(n)}}{\partial \gamma_{l,m}^{(n)}}, \frac{\partial E}{\partial \rho_l^{(n)}} = \frac{\partial E}{\partial x^{(n)}} \frac{\partial x^{(n)}}{\partial \rho_l^{(n)}},$$

where $\frac{\partial E}{\partial x^{(n)}} = \frac{\partial E}{\partial c^{(n)}} \frac{\partial c^{(n)}}{\partial x^{(n)}}$, if $n \leq N_s$, $\quad \frac{\partial E}{\partial x^{(n)}} = \frac{1}{|\Gamma|} \frac{(x^{(n)} - x^{gt})}{\sqrt{\|x^{gt}\|_2^2} \sqrt{\|x^{(n)} - x^{gt}\|_2^2}}$, if $n = N_s + 1$.

The gradients of layer output w.r.t. inputs are computed as $\frac{\partial x^{(n)}}{\partial \beta_l^{(n-1)}}$ and $\frac{\partial x^{(n)}}{\partial z_l^{(n-1)}}$.

# 4    Experiments

We train and test ADMM-Net on brain and chest MR images[2]. For each dataset, we randomly take 100 images for training and 50 images for testing. ADMM-Net is separately learned for each sampling ratio. The reconstruction accuracies are reported as the average NMSE and Peak Signal-to-Noise Ratio (PSNR) over the test images. The sampling pattern in $k$-space is the commonly used pseudo radial sampling. All experiments are performed on a desktop with Intel core i7-4790k CPU.

Table 1: Performance comparisons on brain data with different sampling ratios.

| Method | 20% | | 30% | | 40% | | 50% | | Test time |
|---|---|---|---|---|---|---|---|---|---|
| | NMSE | PSNR | NMSE | PSNR | NMSE | PSNR | NMSE | PSNR | |
| Zero-filling | 0.1700 | 29.96 | 0.1247 | 32.59 | 0.0968 | 34.76 | 0.0770 | 36.73 | 0.0013s |
| TV [2] | 0.0929 | 35.20 | 0.0673 | 37.99 | 0.0534 | 40.00 | 0.0440 | 41.69 | 0.7391s |
| RecPF [4] | 0.0917 | 35.32 | 0.0668 | 38.06 | 0.0533 | 40.03 | 0.0440 | 41.71 | 0.3105s |
| SIDWT | 0.0885 | 35.66 | 0.0620 | 38.72 | 0.0484 | 40.88 | 0.0393 | 42.67 | 7.8637s |
| PBDW [6] | 0.0814 | 36.34 | 0.0627 | 38.64 | 0.0518 | 40.31 | 0.0437 | 41.81 | 35.3637s |
| PANO [10] | 0.0800 | 36.52 | 0.0592 | 39.13 | 0.0477 | 41.01 | 0.0390 | 42.76 | 53.4776s |
| FDLCP [8] | 0.0759 | 36.95 | 0.0592 | 39.13 | 0.0500 | 40.62 | 0.0428 | 42.00 | 52.2220s |
| BM3D-MRI [11] | 0.0674 | 37.98 | 0.0515 | 40.33 | 0.0426 | 41.99 | 0.0359 | 43.47 | 40.9114s |
| Init-Net$_{13}$ | 0.1394 | 31.58 | 0.1225 | 32.71 | 0.1128 | 33.44 | 0.1066 | 33.95 | 0.6914s |
| ADMM-Net$_{13}$ | 0.0752 | 37.01 | 0.0553 | 39.70 | 0.0456 | 41.37 | 0.0395 | 42.62 | 0.6964s |
| ADMM-Net$_{14}$ | 0.0742 | 37.13 | 0.0548 | 39.78 | 0.0448 | 41.54 | 0.0380 | 42.99 | 0.7400s |
| ADMM-Net$_{15}$ | 0.0739 | 37.17 | 0.0544 | 39.84 | 0.0447 | 41.56 | 0.0379 | 43.00 | 0.7911s |

In Tab. 1, we compare our method to conventional compressive sensing MRI methods on brain data. These methods include Zero-filling [22], TV [2], RecPF [4], SIDWT [3], and also the state-of-the-art methods such as PBDW [6], PANO [10], FDLCP [8] and BM3D-MRI [11]. For ADMM-Net, we initialize the filters in each stage to be eight $3 \times 3$ DCT basis (the average DCT basis is discarded). Compared with the baseline methods such as Zero-filling, TV, RecPF and SIDWT, our proposed method produces the best quality with comparable reconstruction speed. Compared with the state-of-the-art methods PBDW, PANO and FDLCP, our ADMM-Net has more accurate reconstruction results with fastest computational speed. For the sampling ratio of 30%, our method (ADMM-Net$_{15}$) outperforms the state-of-the-art methods PANO and FDLCP by 0.71 db. Moreover, our reconstruction speed is around 66 times faster. BM3D-MRI method relies on a well designed BM3D denoiser, it produces higher accuracy, but runs around 50 times slower in computational time than ours. The visual comparisons in Fig. 5 show that the proposed network can preserve the fine image details without obvious artifacts. In Fig. 6(a), we compare the NMSEs and the average test time for different methods using scatter plot. It is easy to observe that our method is the best considering the reconstruction accuracy and running time. Examples of the learned nonlinear functions and the filters are shown in Fig. 7.

Table 2: Comparisons of NMSE and PSNR on chest data with 20% sampling ratio.

| Method | TV | RecPF | PANO | FDLCP | ADMM-Net$_{15}$-B | ADMM-Net$_{15}$ | ADMM-Net$_{17}$ |
|---|---|---|---|---|---|---|---|
| NMSE | 0.1019 | 0.1017 | 0.0858 | 0.0775 | 0.0790 | 0.0775 | 0.0768 |
| PSNR | 35.49 | 35.51 | 37.01 | 37.77 | 37.68 | 37.84 | 37.92 |

*Network generalization ability*: We test the generalization ability of ADMM-Net by applying the learned net from brain data to chest data. Table 2 shows that our net learned from brain data (ADMM-Net$_{15}$-B) still achieves competitive reconstruction accuracy on chest data, resulting in remarkable a generalization ability. This might be due to that the learned filters and nonlinear transforms are performed over local patches, which are repetitive across different organs. Moreover, the ADMM-Net$_{17}$ learned from chest data achieves the better reconstruction accuracy on test chest data.

*Effectiveness of network training*: In Tab. 1, we also present the results of the initialized network for ADMM-Net$_{13}$. As discussed in Section 3.1, this initialized network (Init-Net$_{13}$) is a realization

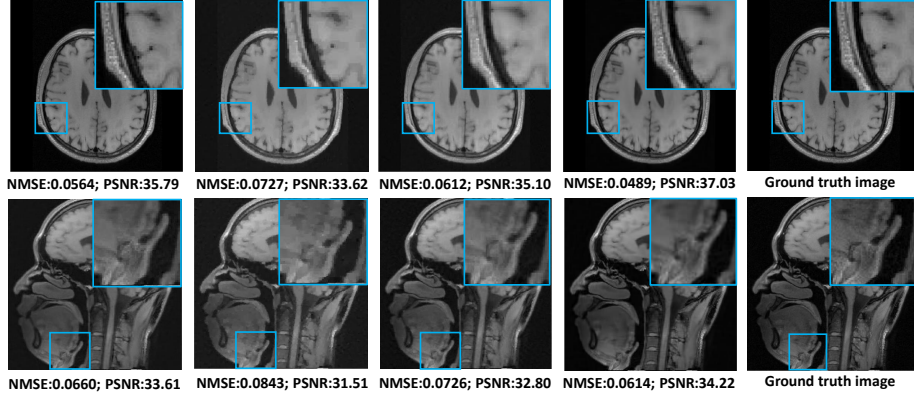

NMSE:0.0564; PSNR:35.79    NMSE:0.0727; PSNR:33.62    NMSE:0.0612; PSNR:35.10    NMSE:0.0489; PSNR:37.03    Ground truth image

NMSE:0.0660; PSNR:33.61    NMSE:0.0843; PSNR:31.51    NMSE:0.0726; PSNR:32.80    NMSE:0.0614; PSNR:34.22    Ground truth image

Figure 5: Examples of reconstruction results with 20% (the first row) and 30% (the second row) sampling ratios. The left four columns show results of ADMM-Net$_{15}$, RecPF, PANO, BM3D-MRI.

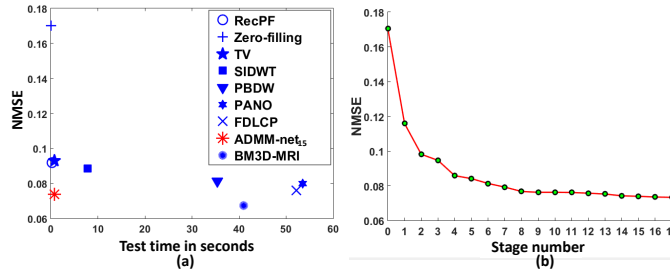

Figure 6: (a) Scatter plot of NMSEs and average test time for different methods; (b) The NMSEs of ADMM-Net using different number of stages (20% sampling ratio for brain data).

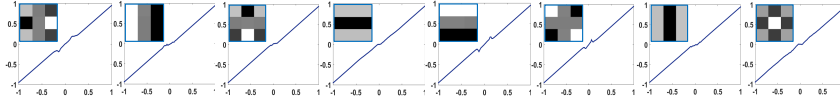

Figure 7: Examples of learned filters in convolution layer and the corresponding nonlinear transforms (the first stage of ADMM-Net$_{15}$ with 20% sampling ratio for brain data).

of the ADMM optimizing Eqn. (12). The network after training produces significantly improved accuracy, e.g., PNSR is increased from 32.71 db to 39.84 db with sampling ratio of 30%.

*Effect of the number of stages*: To test the effect of the number of stages (i.e., $N_s$), we greedily train deeper network by adding one stage at each time. Fig. 6(b) shows the average testing NMSE values using different stages in ADMM-Net under the sampling ratio of 20%. The reconstruction error decreases fast when $N_s < 8$ and marginally decreases when further increasing the number of stages.

*Effect of the filter sizes*: We also train ADMM-Net initialized by two gradient filters with size of $1 \times 3$ and $3 \times 1$ respectively for all convolution and reconstruction layers, the corresponding trained net with 13 stages under 20% sampling ratio achieves NMSE value of 0.0899 and PSNR value of 36.52 db on brain data, compared with 0.0752 and 37.01 db using eight $3 \times 3$ filters as shown in Tab. 1. We also learn ADMM-Net$_{13}$ with 8 filters sized $5 \times 5$ initialized by DCT basis, the performance is not significantly improved, but the training and testing time are significantly longer.

## 5 Conclusions

We proposed a novel deep network for compressive sensing MRI. It is a novel deep architecture defined over a data flow graph determined by an ADMM algorithm. Due to its flexibility in parameter learning, this deep net achieved high reconstruction accuracy while keeping the computational efficiency of the ADMM algorithm. As a general framework, the idea that models an ADMM algorithm as a deep network can be potentially applied to other applications in the future work.

## Footnotes

[1] http://users.eecs.northwestern.edu/~nocedal/lbfgsb.html

[2]CAF Project: `https://masi.vuse.vanderbilt.edu/workshop2013/index.php/Segmentation_Challenge_Details`

[3]Rice Wavelet Toolbox: `http://dsp.rice.edu/software/rice-wavelet-toolbox`

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
