[Supplementary Material · Supp_v1.pdf]

# Deep ADMM-Net for Compressive Sensing MRI
# Supplementary Material

This supplementary material is organized as follows. First, we present the computational procedures of gradients for the ADMM-Net. Second, we include more results and comparisons for MR image reconstruction.

## 1 Gradient Computation for Deep ADMM-Net

Our deep ADMM-Net is defined over the data flow graph shown in Fig. 1. We train the network by gradient-based L-BFGS optimization method. In this section, we present how to compute the gradients using backpropagation. For convenience, we use the matrix operation to substitute the convolution operation in the formulations.

Figure 1: The architecture of deep ADMM-Net with $N_s$ stages. Each stage comprises of reconstruction ($\mathbf{X}$), convolution ($\mathbf{C}$), nonlinear transform ($\mathbf{Z}$), and multiplier update ($\mathbf{M}$) layers. The numbers in the brackets index the stage.

Suppose that the ADMM-Net has $N_s$ stages, followed by a final reconstruction layer $\mathbf{X}^{(\mathbf{N_s}+\mathbf{1})}$. In the training phase of ADMM-Net, the parameters are

$$\Theta = \{(q_{l,i}^{(n)})_{i=1}^{N_c}, D_l^{(n)}, H_L^{(n)}, \rho_l^{(n)}, \eta_l^{(n)}\}_{n=1}^{N_s} \cup \{H_l^{(N_s+1)}, \rho_l^{(N_s+1)}\}, l = 1, \cdots, L,$$

which can be learned over a training set $\Gamma$ by minimizing the following loss function:

$$E(\Theta) = \frac{1}{|\Gamma|} \sum_{(y,x^{gt}) \in \Gamma} \frac{\sqrt{\|\hat{x}(y,\Theta) - x^{gt}\|_2^2}}{\sqrt{\|x^{gt}\|_2^2}}, \tag{1}$$

where $y$ is the sampled data in $k$-space and $x^{gt}$ is the ground-truth. $\hat{x}(y,\Theta)$ is the network output which is just $x^{(N_s+1)}$, i.e., the output of the final reconstruction layer $\mathbf{X}^{(\mathbf{N_s}+\mathbf{1})}$.

The gradients of loss w.r.t. parameters in each layer can be efficiently computed by backpropagation. In the gradient computation using backpropagation, we should first compute the gradient of loss w.r.t. the parameters in the final reconstruction layer $\mathbf{X}^{(\mathbf{N_s}+\mathbf{1})}$ and its inputs. Then we recursively compute the gradients of loss w.r.t. parameters in each stage from stage $N_s$ to 1 backwardly. Within a stage $n$ ($n < N_s + 1$), the gradients are computed in an order of $\mathbf{M}^{(\mathbf{n})}$, $\mathbf{Z}^{(\mathbf{n})}$, $\mathbf{C}^{(\mathbf{n})}$ and $\mathbf{X}^{(\mathbf{n})}$. In

19  such order, we can back propagate the gradients from the loss layer to the first layer $\mathbf{X}^{(1)}$ in stage
20  1 efficiently. Next, we will present how to compute the gradients of loss w.r.t. parameters, and the
21  gradients of layer output(s) to layer input(s) for each layer in the deep architecture.

## 1.1  Multiplier update layer

23  The operation in the multiplier update layer is:

$$\beta_l^{(n)} = \beta_l^{(n-1)} + \eta_l^{(n)}(c_l^{(n)} - z_l^{(n)})(l = 1, 2, \cdots, L). \tag{2}$$

Figure 2: Multiplier update layer

25  As shown in Fig. 2, this layer has three sets of inputs: $\{\beta_l^{(n-1)}\}, \{c_l^{(n)}\}, \{z_l^{(n)}\}$, and the output
26  $\{\beta_l^{(n)}\}$ is the input to compute $\{\beta_l^{(n+1)}\}, \{z_l^{(n+1)}\}$ and $x^{(n+1)}$ in the subsequent layer. The param-
27  eters of this layer are $\eta_l^{(n)}$. The gradients of loss to parameters can be computed as

$$\frac{\partial E}{\partial \eta_l^{(n)}} = \frac{\partial E}{\partial \beta_l^{(n)}} \frac{\partial \beta_l^{(n)}}{\partial \eta_l^{(n)}}, (l = 1, 2, \cdots, L) \tag{3}$$

28  where

$$\begin{aligned}
\frac{\partial E}{\partial \beta_l^{(n)}} &= \frac{\partial E}{\partial \beta_l^{(n+1)}} \frac{\partial \beta_l^{(n+1)}}{\partial \beta_l^{(n)}} + \frac{\partial E}{\partial z_l^{(n+1)}} \frac{\partial z_l^{(n+1)}}{\partial \beta_l^{(n)}} + \frac{\partial E}{\partial x^{(n+1)}} \frac{\partial x^{(n+1)}}{\partial \beta_l^{(n)}}, n \le N_s - 1, \\
\frac{\partial E}{\partial \beta_l^{(n)}} &= \frac{\partial E}{\partial x^{(n+1)}} \frac{\partial x^{(n+1)}}{\partial \beta_l^{(n)}}, n = N_s.
\end{aligned} \tag{4}$$

29  The gradients of the output in this layer w.r.t. parameters are

$$\frac{\partial \beta_l^{(n)}}{\partial \eta_l^{(n)}} = c_l^{(n)} - z_l^{(n)}, (l = 1, 2, \cdots, L) \tag{5}$$

30  and the gradients of the output in this layer w.r.t. the inputs are

$$\frac{\partial \beta_l^{(n)}}{\partial \beta_l^{(n-1)}} = I_{N \times N}, \tag{6}$$

$$\frac{\partial \beta_l^{(n)}}{\partial c_l^{(n)}} = \eta_l^{(n)} I_{N \times N}, \tag{7}$$

$$\frac{\partial \beta_l^{(n)}}{\partial z_l^{(n)}} = -\eta_l^{(n)} I_{N \times N}, \tag{8}$$

33  where $I_{N \times N}$ is an identity matrix sized $N \times N$.

## 1.2 Nonlinear transform layer

The operation in the nonlinear transform layer is:

$$z_l^{(n)} = S_{PLF}(c_l^{(n)} + \beta_l^{(n-1)}; \{p_i, q_{l,i}^n\}_{i=1}^{N_c}), (l = 1, 2, \cdots, L) \tag{9}$$

where $S_{PLF}(\cdot)$ is a piecewise linear function determined by a set of control points $\{p_i, q_{l,i}^n\}_{i=1}^{N_c}$, i.e.

$$z_{u,v,l}^{(n)} = \begin{cases} c_{u,v,l}^{(n)} + \beta_{u,v,l}^{(n-1)} + q_{l,1}^{(n)} - p_1, & c_{u,v,l}^{(n)} + \beta_{u,v,l}^{(n-1)} < p_1, \\ c_{u,v,l}^{(n)} + \beta_{u,v,l}^{(n-1)} + q_{l,N_c}^{(n)} - p_{N_c}, & c_{u,v,l}^{(n)} + \beta_{u,v,l}^{(n-1)} > p_{N_c}, \\ q_{l,k}^{(n)} + \frac{(c_{u,v,l}^{(n)} + \beta_{u,v,l}^{(n-1)} - p_k)(q_{l,k+1}^{(n)} - q_{l,k}^{(n)})}{p_2 - p_1}, & p_1 \le c_{u,v,l}^{(n)} + \beta_{u,v,l}^{(n-1)} \le p_{N_c}, \end{cases} \tag{10}$$

where $k = \lfloor \frac{c_{u,v,l}^{(n)} + \beta_{u,v,l}^{(n-1)} - p_1}{p_2 - p_1} \rfloor$, $(u, v)$ indicates the coordinate of a pixel within image region $\Omega$.

Figure 3: Nonlinear transform layer

As shown in Fig. 3, this layer has two sets of inputs: $\{\beta_l^{(n-1)}\}$, $\{c_l^{(n)}\}$, and the output $\{z_l^{(n)}\}$ is the input to compute $\{\beta_l^{(n)}\}$, $x^{(n+1)}$ in the subsequent layer. The parameters are $\{q_{l,i}^{(n)}\}_{i=1}^{N_c}$ related to the predefined positions $\{p_i\}_{i=1}^{N_c}$, the gradients of the loss w.r.t. parameters are

$$\frac{\partial E}{\partial q_{l,i}^{(n)}} = \frac{\partial E}{\partial z_l^{(n)}} \frac{\partial z_l^{(n)}}{\partial q_{l,i}^{(n)}}, (l = 1, 2, \cdots, l) \tag{11}$$

where

$$\frac{\partial E}{\partial z_l^{(n)}} = \frac{\partial E}{\partial \beta_l^{(n)}} \frac{\partial \beta_l^{(n)}}{\partial z_l^{(n)}} + \frac{\partial E}{\partial x^{(n+1)}} \frac{\partial x^{(n+1)}}{\partial z_l^{(n)}}, n \le N_s. \tag{12}$$

We compute the gradients of the output in this layer w.r.t. parameter as

$$\frac{\partial z_{u,v,l}^{(n)}}{\partial q_{l,k}^{(n)}} = \begin{cases} 0, & c_{u,v,l}^{(n)} + \beta_{u,v,l}^{(n-1)} < p_1, \\ 0, & c_{u,v,l}^{(n)} + \beta_{u,v,l}^{(n-1)} > p_{N_c}, \\ 1 - \frac{c_{u,v,l}^{(n)} + \beta_{u,v,l}^{(n-1)} - p_k}{p_2 - p_1}, & p_1 \le c_{u,v,l}^{(n)} + \beta_{u,v,l}^{(n-1)} \le p_{N_c}, \end{cases} \tag{13}$$

and the gradients of the output in this layer w.r.t. the inputs as

$$\frac{\partial z_{u,v,l}^{(n)}}{\partial \beta_{u,v,l}^{(n-1)}} = \begin{cases} 1, & c_{u,v,l}^{(n)} + \beta_{u,v,l}^{(n-1)} < p_1, \\ 1, & c_{u,v,l}^{(n)} + \beta_{u,v,l}^{(n-1)} > p_{N_c}, \\ \frac{q_{l,k+1}^{(n)} - q_{l,k}^{(n)}}{p_2 - p_1}, & p_1 \le c_{u,v,l}^{(n)} + \beta_{u,v,l}^{(n-1)} \le p_{N_c}, \end{cases} \tag{14}$$

where $k = \lfloor \frac{c_{u,v,l}^{(n)} + \beta_{u,v,l}^{(n-1)} - p_1}{p_2 - p_1} \rfloor$, $l = 1, 2, \cdots, L$ and $(u, v) \in \Omega$.

$$\frac{\partial z_{u,v,l}^{(n)}}{\partial c_{u,v,l}^{(n)}} = \frac{\partial z_{u,v,l}^{(n)}}{\partial \beta_{u,v,l}^{(n-1)}}. \tag{15}$$

Figure 4: Convolution layer

## 1.3 Convolution layer

The operation in the convolution layer is:

$$c_l^{(n)} = D_l^{(n)} x^{(n)} (l = 1, 2, \cdots, L). \tag{16}$$

As shown in Fig. 4, this layer has an input: $x^{(n)}$, and the output $\{c_l^{(n)}\}$ is the input to compute $\{\beta_l^{(n)}\}, \{z_l^{(n)}\}$ in the subsequent layer. The parameters are $D_l^{(n)}$, let the filter $D_l^{(n)} = \sum_{m=1}^{t} \omega_{l,m}^{(n)} B_m$, where $B_m$ is a DCT basis, and $\omega_{l,m}^{(n)}$ is the related filter coefficient. We update the filters by computing the gradients w.r.t. filters coefficients, which are

$$\frac{\partial E}{\partial \omega_{l,m}^{(n)}} = \frac{\partial E}{\partial c_l^{(n)}} \frac{\partial c_l^{(n)}}{\partial \omega_{l,m}^{(n)}}, (l = 1, 2, \cdots, L) \tag{17}$$

where

$$\frac{\partial E}{\partial c_l^{(n)}} = \frac{\partial E}{\partial \beta_l^{(n)}} \frac{\partial \beta_l^{(n)}}{\partial c_l^{(n)}} + \frac{\partial E}{\partial z_l^{(n)}} \frac{\partial z_l^{(n)}}{\partial c_l^{(n)}}, n \leq N_s. \tag{18}$$

We compute the gradients of the output in this layer w.r.t. parameters

$$\frac{\partial c^{(n)}}{\partial \omega_{l,m}^{(n)}} = B_m x^{(n)}, (l = 1, 2, \cdots, L) \tag{19}$$

and the gradients w.r.t. the output in this layer are

$$\frac{\partial c_l^{(n)}}{\partial x^{(n)}} = D_l^{(n)} I_{N \times N}, (l = 1, 2, \cdots, L). \tag{20}$$

## 1.4 Reconstruction layer

The operation in the reconstruction layer is:

$$x^{(n)} = F^T (P^T P + \sum_{l=1}^{L} \rho_l^{(n)} F H_l^{(n)T} H_l^{(n)} F^T)^{-1} [P^T y + \sum_{l=1}^{L} \rho_l^{(n)} F H_l^{(n)T} (z_l^{(n-1)} - \beta_l^{(n-1)})]. \tag{21}$$

As shown in Fig. 5, this layer has two sets of inputs: $\{z_l^{(n-1)}\}$ and $\{\beta_l^{(n-1)}\}$, and the output $x^{(n)}$ is the input to compute $\{c_l^{(n)}\}$ in the subsequent layer. The parameters of this layer are $H_l^{(n)}, \rho_l^{(n)}$. Similar to convolution layer, we represent the filter by $H_l^{(n)} = \sum_{m=1}^{s} \gamma_{l,m}^{(n)} B_m$, where $\{\gamma_{l,m}^{(n)}\}$ is a set of filter coefficients to be learned. The gradients of loss w.r.t. parameters are computed as

$$\frac{\partial E}{\partial \gamma_{l,m}^{(n)}} = \frac{\partial E}{\partial x^{(n)}} \frac{\partial x^{(n)}}{\partial \gamma_{l,m}^{(n)}}, \tag{22}$$

$$\frac{\partial E}{\partial \rho_l^{(n)}} = \frac{\partial E}{\partial x^{(n)}} \frac{\partial x^{(n)}}{\partial \rho_l^{(n)}}, \tag{23}$$

Figure 5: Reconstruction layer

where

$$\frac{\partial E}{\partial x^{(n)}} = \sum_{l=1}^{L} \frac{\partial E}{\partial c_l^{(n)}} \frac{\partial c_l^{(n)}}{\partial x^{(n)}}, n \le N_s, \tag{24}$$

$$\frac{\partial E}{\partial x^{(n)}} = \frac{1}{|\Gamma|} \frac{(x^{(n)} - x^{gt})}{\sqrt{\|x^{gt}\|_2^2}\sqrt{\|x^{(n)} - x^{gt}\|_2^2}}, n = N_s + 1. \tag{25}$$

Then, we compute the gradients of the output in this layer w.r.t. parameters

$$
\begin{aligned}
\frac{\partial x^{(n)}}{\partial \gamma_{l,m}^{(n)}} &= F^T\{\frac{\partial Q}{\partial \gamma_{l,m}^{(n)}}[P^T y + \sum_{l=1}^{L} \rho_l^{(n)} F H_l^{(n)T}(z_l^{(n-1)} - \beta_l^{(n-1)})] \\
&\quad + Q\rho_l^{(n)} F \frac{\partial H_l^{(n)T}(z_l^{(n-1)} - \beta_l^{(n-1)})}{\partial \gamma_{l,m}^{(n)}}\} \\
&= -\rho_l^{(n)} F^T\{Q^2 \frac{\partial F H_l^{(n)T} H_l^{(n)} F^T}{\partial \gamma_{l,m}^{(n)}}[P^T y + \sum_{l=1}^{L} \rho_l^{(n)} F H_l^{(n)T}(z_l^{(n-1)} - \beta_l^{(n-1)})] \\
&\quad - QFB_m^T(z_l^{(n-1)} - \beta_l^{(n-1)})\} \\
&= -\rho_l^{(n)} F^T\{Q^2[FB_m^T H_l^{(n)} F^T + F H_l^{(n)T} B_m F^T][P^T y + \sum_{l=1}^{L} \rho_l^{(n)} F H_l^{(n)T} \\
&\quad (z_l^{(n-1)} - \beta_l^{(n-1)})] - QFB_m^T(z_l^{(n-1)} - \beta_l^{(n-1)})\}, \tag{26}
\end{aligned}
$$

$$
\begin{aligned}
\frac{\partial x^{(n)}}{\partial \rho_l^{(n)}} &= F^T\{\frac{\partial Q}{\partial \rho_l^{(n)}}[P^T y + \sum_{l=1}^{L} \rho_l^{(n)} F H_l^{(n)T}(z_l^{(n-1)} - \beta_l^{(n-1)})] \\
&\quad + QF H_l^{(n)T}(z_l^{(n-1)} - \beta_l^{(n-1)})\} \\
&= -F^T\{Q^2(FH_l^{(n)T} H_l^{(n)} F^T)[P^T y + \sum_{l=1}^{L} \rho_l^{(n)} F H_l^{(n)T}(z_l^{(n-1)} - \beta_l^{(n-1)})] \\
&\quad - QF H_l^{(n)T}(z_l^{(n-1)} - \beta_l^{(n-1)})\}, \tag{27}
\end{aligned}
$$

where

$$Q = [P^T P + \sum_{l=1}^{L} \rho_l^{(n)} F H_l^{(n)T} H_l^{(n)} F^T]^{-1}. \tag{28}$$

The gradients of the output in this layer w.r.t. the inputs are

$$\frac{\partial x^{(n)}}{\partial \beta_l^{(n-1)}} = -\rho_l^{(n)} F^T Q F H_l^{(n)T} I_{N \times N}, \tag{29}$$

$$\frac{\partial x^{(n)}}{\partial z_l^{(n-1)}} = \rho_l^{(n)} F^T Q F H_l^{(n)T} I_{N \times N}. \tag{30}$$

 ## 2 More Results

Figure 6: Brain MR images reconstruction using different methods with sampling ratio of 20%. (a) ADMM-Net$_{15}$; (b) RecPF; (c) PANO; (d) Ground truth. The average CPU time of (a)-(c) are 0.797s, 0.326s and 48.683s.

Figure 7: Brain MR images reconstruction using different methods with sampling ratio of 30%. (a) ADMM-Net$_{15}$; (b) RecPF; (c) PANO; (d) Ground truth. The average CPU time of (a)-(c) are 0.792s, 0.311s and 47.818s.

NMSE:0.0343
PSNR :40.2860

NMSE:0.0338
PSNR :40.9815

NMSE:0.0549
PSNR :36.1159

**(a)**

**ADMM-Net**
**0.794s**

NMSE:0.0414
PSNR :38.6531

NMSE:0.0409
PSNR :39.3339

NMSE:0.0662
PSNR :34.5070

**(b)**

**RecPF**
**0.323s**

NMSE:0.0373
PSNR :39.5577

NMSE:0.0359
PSNR :40.4686

NMSE:0.0584
PSNR :35.6021

**(c)**

**PANO**
**48.759s**

**(d)**

**Ground**
**truth**

Figure 8: Brain MR images reconstruction using different methods with sampling ratio of 40%. (a) ADMM-Net$_{15}$; (b) RecPF; (c) PANO; (d) Ground truth. The average CPU time of (a)-(c) are 0.794s, 0.323s and 48.759s.

Figure 9: Brain MR images reconstruction using different methods with sampling ratio of 50%. (a) ADMM-Net$_{15}$; (b) RecPF; (c) PANO; (d) Ground truth. The average CPU time of (a)-(c) are 0.797s, 0.320s and 48.875s.

Figure 10: Chest MR images reconstruction using different methods with sampling ratio of 20%.
(a) ADMM-Net$_{17}$; (b) RecPF; (c) FDLCP; (d) Ground truth. The average CPU time of (a)-(c) are
0.888s, 0.367s and 54.623s.