[Reviews · NeurIPS 2016]

Reviewer 1

Summary

The paper addresses the problem of compressive sensing MRI (CS-MRI) by proposing a "deep unfolding" approach (cf. http://arxiv.org/abs/1409.2574) with a sparsity-based data prior and inference via ADMM. All layers of the proposed ADMM-Net are based on a generalization of ADMM inference steps and are discriminatively trained to minimize a reconstruction error. In contrast to other methods for CS-MRI, the proposed approach offers both high reconstruction quality and fast run-time.

Qualitative Assessment

The experimental results appear to very good compared to other CS-MRI approaches, since the proposed ADMM-Net offers both high quality and fast runtime, whereas the competing methods either have only fast runtime or good reconstruction quality. (The chosen methods for comparison seem to be relatively recent, but I don't know whether they are state-of-the-art for CS-MRI.) I think generalization of the learned network is an important point that deserves more emphasis. It is only briefly mentioned (l. 210-215) that the network trained on the brain dataset also works well on the chest dataset. Would this kind of generalization ability be useful in practice if this were to be used with an actual MRI machine? Overall, the proposed unrolled inference approach is similar to [19], which uses a similar data prior with similar inference (but without Lagrange multipliers) for image denoising and deconvolution. I agree with the paper and also think that this is the first time that unrolled inference has been applied to ADMM, especially in the context of CS-MRI. However, it seems that the paper adopts several design decisions from [19] (and/or earlier papers) that should be acknowledged in the text (such as the use of L-BFGS for training, DCT basis for filter coefficients, size and number of filters). Especially the general concept of unrolled/unfolded inference has been used before (in [19] and earlier, also see http://arxiv.org/abs/1409.2574), which should definitely be acknowledged. Miscellaneous: - Line 74: Please cite a reference for ADMM (maybe a book). - Fig 2: Piecewise linear functions don't look differentiable (the supplemental material doesn't talk about this). Why is this not a problem? - It is interesting that performance only saturates after around 15 stages. This is in contrast to [19], where performance saturates after only 4-5 stages. Also, it would be interesting to understand why larger filters don't lead to improved results. - Have all stages / network layers been trained jointly with L-BFGS? Maybe greedy training (for each stage) does work better (as in [19])? - Many equations (e.g. Eq. 10) contain "sqrt(norm_2^2(...))", which simply is "norm_2(...)". - Have any constraints been imposed on model parameters during learning? - A comment regarding line 161 ("It is challenging to compute the gradients of loss w.r.t. parameters using backpropagation over the deep architecture in Fig. 1, because it is a directed graph instead of a chain structure.") I basically have to disagree, since the chain rule can still be applied. Also, common deep learning frameworks even take care of this automatically as long as each kind of layer is implemented properly.

Confidence in this Review

2-Confident (read it all; understood it all reasonably well)


Reviewer 2

Summary

The authors present a deep architecture that implements an ADMM for compressed sensing applied to MRI recovery. The ADMM algorithm has proven to be useful for solving problems with differentiable and non-differentiable terms, and therefore has a clear link with compressed sensing. Experiments prove some gain in performance with respect to the state of the art, specially in terms of computational cost at test time.

Qualitative Assessment

The idea of reimplementing an iterative algorithm in a deep architecture is not new, but solving a CS problem for MRI recovery with a Deep-ADMM architecture is novel as far as I can say. I am a bit worried about the correspondence between the "classical" and the "deep" learning problems. Indeed, the classical problem is very well stated as minimizing the augmented Lagrangian. However, when discussing the implementation of the convolutional layer, the authors say that they increase the network capacity by decoupling the transform matrix of the convolutional from the one of the reconstruction layer. Therefore, we are not solving for the augmented Lagrangian anymore, but for something else. What is the interpretation in terms of "classical" learning? Can you rewrite the optimization problem you are actually solving for and discuss similarities and differenceswith the original augmented Lagrangian? The authors claim something before equation (8) that is not clear to me and would require explanation or proof. When Implementing the non-linear transform layer (I.e. the shrinkage), how do you know that using a piecewise linear function is more general than trying to implement the shrinkage. I mean, I understand that fitting a piecewise linear function can approximate many other functions and therefore has some flexibility: but can it implement any shrinkage operator with piecewise linear functions (recall that the control points are predefined). My understanding is that even if more flexible, this layer is perhaps not more general than the original shrinkage. When writing the gradients used in the back-propagation, I'd rather see in the main text the actual gradients and not general chain rules that do not provide any insight on the effects of each back-propagation step. The experimental results should be sharpened-up. It is not really convincing to me that ADMM-NET-14 does not reach state-of-the-art performance, but ADMM-NET-15 does. It is very intuitive, since you get one iteration more of the ADMM. But how can I choose the number of iterations a priori? In regular ADMM different convergence criteria can be evaluated, but with your approach the number of iterations has to be fixed in advance. Importantly, this is critical, because the number of iterations defines the number of layers, and therefore the amount of parameters of the network. Therefore, while for classical ADMM, the number of parameters is fixed with the number of iterations, here it grows. Therefore, the number of data must grow with the number of iterations. Can you please comment on that and provide some insights? In this topic, I would also appreciate to see that the parameters also converge (as the MMSE does in Fig. 6b). Typos: - Generally speaking, check the usage of the definite article "the" very carefully. -P1L19 techniques are -P1L22 CS-MRI (without the) -P1L29 uses groupS -P1L36 proven (not proved) -P2L56 non-local techniqueS -P1L58 propose (without d)

Confidence in this Review

2-Confident (read it all; understood it all reasonably well)


Reviewer 3

Summary

This paper interpret the ADMM algorithm originally designed to solve the optimization problem (1) as a deep network. The learnable parameters of the the network include the parameters of ADMM, the filters in the sparsifying transformations and the shrinkage operators. Authors used SATA dataset and compared with some state-of-the-art CS algorithms. The proposed method outperform the those algorithm in both PSNR values and computation time.

Qualitative Assessment

Here are some questions and comments: 1) As we know deep learning methods, especially when there are many parameters to learn, require a large training set. The authors only used 100 images as testing. I suggest authors elaborate on the total number of parameters (at least roughly) needed to be trained, and why merely 100 test images is enough. It is also important for the readers to understand why the proposed algorithm is so fast (under 1 sec in average, which from the writing includes both training and test?) 2) The benchmark regularizer for image restoration is the well-known BM3D. It is a very fast regularizer which utilizes both local and nonlocal sparsity of an image. It is NOT learning based method, thus works when there isn't a reliable data set to learn from. Hence, it is more flexible. I suggest the authors compare with BM3D (for example: [Eksioglu, Ender M. "Decoupled Algorithm for MRI Reconstruction Using Nonlocal Block Matching Model: BM3D-MRI." Journal of Mathematical Imaging and Vision (2016): 1-11]). All the methods the authors compared with in the manuscript are known to be less effective than BM3D in general. 3) How does the algorithm scale w.r.t. the dimension of the image? The major computation issue with ADMM is memory issue for 3D image data. For this reason, people are trying to avoid saving the coefficients (variable z in the paper) explicitly, especially when the transforms {D_l} is redundant. Poor scalability may limit the application of the proposed method.

Confidence in this Review

2-Confident (read it all; understood it all reasonably well)


Reviewer 4

Summary

In this paper, the authors unfolded ADMM framework for compressed sensing MRI into a deep network in which the network parameters such as filters, regularization parameters, shrinkages etc can be learned from the training data. In order to derive the backpropagtiaon based training equation, the authors derived a data flow graph and the associated back-propagation algorithms. The authors then conducted the extensive numerical experiments to demonstrate that the proposed ADMM-net outperforms the existing approaches in terms of complexity and reconstruction quality.

Qualitative Assessment

The idea of unfolding of sparse recovery framework to deep network is not new. However, to my best knowledge, the proposed unfolding scheme of ADMM framework and the associated training scheme are quite novel. Therefore, I strongly recommend this paper. However, there are several issues that should be addressed before final acceptance. - To my understanding, the data set from CAF project are MR images, which are real valued. Then, the k-space data for the training and validation should be different from the actual acquisition scenario, because the actual MR images is complex-valued. - In terms of visual quality, there were not much differences between the proposed method and the existing approaches. - In the current setup, many parameters should be learned. However, the learned shrinkage operator appears similar to the standard shrinkage operator. Is this critical ? Comparison without shrinkage learning would be helpful for readers. - There are recent CS-MRI approaches using transform learning or co-sparse dictionary learning. The current scheme has similarity to such schemes even though in the existing approaches the learning is done on-line rather than using training data. The differences should be explained.

Confidence in this Review

3-Expert (read the paper in detail, know the area, quite certain of my opinion)


Reviewer 5

Summary

This paper provides a deep network structure for compressive sensing MRI image reconstruction. The proposed network is inspired by the ADMM iteration of solving the objective of image restoration, and is called ADMM-Net. ADMM-Net automates the selection of transformation, functions and parameters by training, and enjoys more flexibility by learning those variables. ADMM-Net outperforms some recent image restoration methods such as dictionary learning (Zhan et al.) and non-local operator (Qu et al.).

Qualitative Assessment

This paper uses the data flow graph of ADMM to construct deep network for MRI image restoration, which I considered as novel and interesting. However, based on my expertise of DNN, I am not quite confident about how original and how important the idea of using the iteration steps of optimization to guide the construction of network is. I am willing to upgrade the novelty score if more discussion on DNN literature is provided. Moreover, a discussion of what optimization method for what applications can be used to construct DNN will be interesting. I am willing to upgrade the impact score if sufficient evidence is provided that the proposed method is not only interesting for MRI researchers. Generally, the paper is well organized and I enjoyed the reading. Thanks a lot for submitting the nice work. Some other issues I would like the authors to address, 1. The dual variable update in Eq.(4) seems incorrect. And there is no closed form solution for this subproblem in every iteration. The update of dual variable in ADMM is dual ascent. 2. line 115, the author claims D and H are different to increase network capacity. What will happen if constraint D=H? It seems significantly harder to solve, but will it harm the performance? 3. How many parameters in total does the ADMM-Net have? Are there any techniques used to prevent overfitting? Also, the experiment seems to suggest the more layers, the better? 4. I am worried about the fairness of the experiments. Did the major baselines [11][13] also use 100 images for training? The objective/loss of ADMM-Net is MSE and the evaluation is also MSE. There is no comparison with other NN methods. I am interested in the performance of a simple CNN with MSE as loss. Some minor issues, 5. in the introduction part, the data flow graph explanation of the proposed method is a little confusing; 6. line 60-61, it is unclear what 'this approach' refers to, ADMM-Net? How justified? By experiments? 7. line 145, why L-BFGS, not SGD that is widely used in NN?

Confidence in this Review

2-Confident (read it all; understood it all reasonably well)


Reviewer 6

Summary

This paper proposed a compressive sensing based MRI reconstruction algorithm using neural network. The basic idea is to convert the convention optimization based CS reconstruction algorithm into a fixed neural network learned with back-propagation algorithm. Specifically, the ADMM-based CS reconstruction is approximated with a deep neural network. Experimental results show that the approximated neural network outperforms several existing CS-MRI algorithms with less computational time.

Qualitative Assessment

Overall, this is a good paper with interesting results. However, some technique details of the algorithms are missing. For example, in reconstruction operation of Eq. (5), the matrix to be inverted is very large. How is X^{n} computed in both forward and backward computation? It is stated that a piecewise linear function is used instead of hard/soft thresholding function. Please show the specific form of the piecewise linear function.

Confidence in this Review

2-Confident (read it all; understood it all reasonably well)